# Pharmacokinetic Herb-Drug Interactions of Glipizide with *Andrographis paniculata* (Burm. f.) and Andrographolide in Normal and Diabetic Rats by Validated HPLC Method

**DOI:** 10.3390/molecules27206901

**Published:** 2022-10-14

**Authors:** Elza Sundhani, Agung Endro Nugroho, Arief Nurrochmad, Ika Puspitasari, Dita Amalia Prihati, Endang Lukitaningsih

**Affiliations:** 1Faculty of Pharmacy, Universitas Gadjah Mada, Sekip Utara, Yogyakarta 55281, Indonesia; 2Department of Pharmacology and Clinical Pharmacy, Faculty of Pharmacy, Universitas Muhammadiyah Purwokerto, Jl. KH. Ahmad Dahlan Dukuhwaluh, Purwokerto 53182, Indonesia; 3Department of Pharmacology and Clinical Pharmacy, Faculty of Pharmacy, Universitas Gadjah Mada, Sekip Utara, Yogyakarta 55281, Indonesia; 4Laboratory of Advanced Pharmaceutical Sciences, Faculty of Pharmacy, Universitas Gadjah Mada, Sekip Utara, Yogyakarta 55281, Indonesia; 5Department of Pharmaceutical Chemistry, Faculty of Pharmacy, Universitas Gadjah Mada, Sekip Utara, Yogyakarta 55281, Indonesia

**Keywords:** herb–drug interactions, *Andrographis paniculata*, andrographolide, glipizide

## Abstract

Co-administered medicinal herbs can modify a drug’s pharmacokinetics (PK), effectiveness, and toxicity. *Andrographis paniculata* (Burm. f.) ethanolic extract (APE) and andrographolide (AND) (a potent CYP2C9 inducer/inhibitor) can alter the pharmacokinetic parameters of glipizide (GLZ). This study aimed to determine the potential pharmacokinetics of herb–drug interactions between GLZ and APE/AND in the plasma of normal and diabetic rats using the HPLC bioanalysis method. The glipizide bioanalytical method established with RP-HPLC/UV instrument was validated following the EMA guidelines. GLZ was administered alone and in combination with APE or AND to normal and diabetic rats. The GLZ pharmacokinetic parameters were estimated according to the correlation between concentration and sampling time using the PK solver program. A simple and rapid GLZ bioanalysis technique with a lower limit of quantitation of 25 ng/mL was developed and presented the following parameters: accuracy (error ≤ 15%), precision (CV ≤ 15%), selectivity, stability, and linearity (R^2^ = 0.998) at concentrations ranging 25–1500 ng/mL. APE administration significantly improved the C_max_ and AUC_0–t_/AUC_0–∞_ GLZ values in normal and diabetic rats (*p* < 0.05). AND significantly reduced the bioavailability of GLZ in diabetic rats with small values of T 1/2, C_max_, and AUC_0–t_/AUC_0__–∞_ (*p* < 0.05). This combination can be considered in administering medications because it can influence the pharmacological effects of GLZ.

## 1. Introduction

Complementary and alternative medicine (CAM) derived from natural products or plant-based substances has been increasingly utilized to treat diseases. Its application is expanding in Asian nations, the United States, Europe, and Australia [1,2,3]. Diabetes mellitus is a disease that responds favorably to CAM treatments, particularly plant-based natural products [3,4]. Approximately 78% of patients with diabetic mellitus use herbal medicines and supplements as an alternative treatment [5,6].

According to the International Diabetes Federation, diabetes mellitus affects more than 500 million people worldwide (10.5% of the world’s adult population) [7]. The prevalence of diabetes mellitus influences the production of many herbal medicines as alternative treatments. The components of herbal medicinal compounds can cause herb–drug interactions (HDIs) because they can act as reversible inhibitors, irreversible inhibitors, or inductors of CYP450 enzymes or cause synergistic, additive, or antagonistic effects on the pharmacological activity of drugs [8]. The simultaneous usage of medicines and natural compounds can lead to pharmacokinetic (absorption, distribution, metabolism, and elimination) and pharmacodynamic (pharmacological activity) interactions [9,10,11].

Antidiabetic drugs, including sulfonylureas, meglitinide analogs, thiazolidinediones, and DPP4 inhibitors, are metabolized mainly by the enzymes CYP2C8, CYP2C9, CYP3A4, and CYP2C19 [12]. CYP2C9 is the most common member of the P450 2C (CYP2C) subfamily and has a role in the metabolism of oral hypoglycemic agents. This enzyme is also implicated in the metabolism of numerous anticoagulant drugs, NSAIDs, and antihypertensive drugs [13,14]. Glipizide is a sulfonylurea antidiabetic medication metabolized by CYP2C9 and CYP2C19 [15,16,17] to inactive metabolites in 3-cis-hydroxyglipizide (15%) and 4-trans-hydroxyglipizide (71%) [18]. Alterations in glipizide plasma concentrations caused by metabolic phase variations can affect the therapy’s effectiveness. An example is HDIs that occur when combining drugs with natural substances that have inducing or inhibiting effects on their metabolizing enzymes.

*Andrographis paniculata* (Burm. f.) or Sambiloto is one of the herbs commonly used as alternative medicine due to its pharmacological activities, including antidiabetic, anti-inflammatory, anti-obesity, antioxidant, and anti-dengue properties [19,20,21,22]. In addition to its potential pharmacological efficacy, *A. paniculata* can produce HDIs with several conventional drugs. *A. paniculata*’s secondary metabolites, including andrographolide (AND), 14-deoxy-11,12-didehydroandrographolide, and andrographidine A, exhibit stable affinity and binding to receptors that play a role in the expression of CYP450 metabolizing enzymes, such as constitutive androstane receptor and pregnane X receptor (PXR) [23]. *A. paniculata* and its major secondary metabolite, andrographolide, can inhibit the kinetics of several enzymes, including CYP2E1, and decrease the expression of CYP2C9 and CYP3A proteins on human hepatic cytochrome and Caco-2 cell line [24,25,26,27].

*A. paniculata* ethanolic extract (APE) and AND can alter the pharmacokinetic profile of a drug by increasing the clearance (CL) value and significantly decreasing the AUC of theophylline, etoricoxib, nabumetone, and naproxen [28,29,30,31]. APE can also increase the AUC and reduce the CL of gliclazide and midazolam [32,33]. AND, the primary component of APE, can alter the pharmacokinetic parameters of aminophylline, doxofylline, meloxicam, glyburide, glimepiride, metformin, and warfarin by enhancing their AUC and Tmax values and significantly reducing their CL values [34,35,36,37,38].

APE and AND exhibit HDIs with several drugs; however, their pharmacokinetic interactions with the antidiabetic agent glipizide have never been examined. Drug interaction investigations, particularly during the pharmacokinetic phase, require a selective, precise, and efficient method for monitoring plasma glipizide concentration. The chromatographic approach is the most commonly used because it can obtain low detection limits, produce structural information, and analyze analytes with varying polarities [39]. LC–MS/MS and RP-HPLC/UV have been applied extensively to assess glipizide in the plasma matrix. Compared with LC–MS/MS, RP-HPLC/UV is simpler in terms of sample preparation and less expensive. Prior to the analysis of plasma glipizide concentration in normal and diabetic rats, the analytical method was verified following the European Medicines Agency (EMA) guidelines. The best and most valid analytical method was applied to examine the pharmacokinetics of glipizide in plasma from the rats co-administered with APE and AND.

## 2. Materials and Methods

### 2.1. Chemicals and Reagents

Glipizide of pharmaceutical secondary standard (Sigma–Aldrich, St. Louis, MO, USA), andrographolide (98%, Sigma–Aldrich), simplicia *Andrographis paniculata* (Burm. f.) collection from B2P2TOOT Tawangmangu, Central Java, Indonesia, acetonitrile of HPLC grade (Smartlab), methanol of HPLC grade, potassium dihydrogen phosphate (KH_2_PO_4_), distilled water (PT. Brataco, Indonesia), and blank plasma were used in this study.

### 2.2. HPLC Condition

A set of HPLC instruments (Hitachi UV-vis L-2420 detector [at 233 nm], Hitachi L-2130 HPLC pump, D-2000 HSM elite software, Phenomenex Luna^®^ 5 m C18 100 chromatographic column, LC column 250 mm × 4.6 mm) were employed in this work. The mobile phase consisted of KH_2_PO_4_ and acetonitrile (57%:43%) and had a pH of 4.25 and a flow rate of 1 mL/min. The mobile phase was filtered using a 0.45 m pore filter and degassed using a bath sonicator before use.

### 2.3. Preparation of Standard Stock and Working Solutions

In brief, 10 mg of glipizide was weighed and transferred to a 10 mL volumetric flask before methanol was added to the mark (Solution A, glipizide 1000 mg/mL). Afterward, 1 mL of solution A was pipetted into a 10 mL volumetric flask, methanol was added to the calibration mark, and the mixture was homogenized (Solution B, glipizide 100 g/mL). Solution B was pipetted into a 10 mL flask, and methanol was added to the mark to create a series of working standard solutions (0.25–15 μg/mL).

### 2.4. Preparation of Spiked Plasma

In brief, 200 µL of plasma was administered with a standard solution of glipizide in a centrifuge tube. Deprotonation was conducted by adding 1000 µL of acetonitrile and centrifuging at 14,000 rpm at 40 °C for 10 min. The supernatant was extracted, transferred to a vial (twofold replication), and dried. After drying, 1 mL of mobile phase was added to the residue, which was then vortexed for 1 min, filtered over a 0.45 µm nylon/PVDF membrane, and introduced into the HPLC instrument.

### 2.5. System Suitability Test (SST)

SST was conducted to measure the amount of glipizide in plasma using a standard solution of glipizide with a concentration of 1500 ng/mL. The SST criteria for the percentage RSD area of glipizide in standard solutions for six injection replications (≤2.0%, tailings factor [asymmetry] ≤ 2.0, and theoretical plate [N] *≥* 2000) were satisfied.

### 2.6. Method Validation

The bioanalytical method for measuring glipizide in plasma was validated following the EMA guidelines. The validation procedure consisted of selectivity, calibration curve, lower limit of quantitation (LLOQ), accuracy, and precision, carryover, dilution integrity, and stability.

#### 2.6.1. Selectivity and Carryover

Selectivity test was conducted by injecting plasma for interference with glipizide peaks. Selectivity is achieved when the peak interference in the retention time of glipizide in blank plasma is ≤20% of the LLOQ of glipizide. For carryover, blank plasma was injected into the HPLC system after the injection of the highest concentration plasma spike.

#### 2.6.2. Calibration Curve and LLOQ

Calibration standards were prepared by adding standard glipizide solution to 200 μL (spike) of blank plasma to obtain the final serial concentrations of 25, 75, 150, 300, 550, 700, 950, 1150, 1250, and 1500 ng/mL. The values of linear regression, slope, intercept, and percentage recovery were calculated from the concentration versus area correlation. The calibration standard satisfies the parameters when the difference between the observed concentration of the standard solution and its theoretical concentration is less than ±15%, except for LLOQ for which the difference cannot exceed ±20%. A minimum of six series of standard concentrations must comprise the calibration curve, and 75% of the standard solutions must meet the criterion. LLOQ was determined by injecting standard glipizide solution into plasma at concentrations of 15, 25, and 30 ng/mL to obtain six replicates. LLOQ was determined at the measured concentration with a percentage difference of ±20%.

#### 2.6.3. Accuracy and Precision

Glipizide standard solution was added to the plasma at four concentration levels within the calibration curve range: LLOQ, 3 × LLOQ (low QC), 30% to 50% of the calibration curve range (medium QC), and at least 75% of the highest calibration curve (high QC). Accuracy is satisfied when the difference between the average value of the measured QC sample concentration and the theoretical concentration is less than ±15%, except for LLOQ for which the difference must not exceed ± 20%. Precision is satisfied when the standard deviation value of the QC sample concentration measured from six replications did not exceed 15%, except for LLOQ for which the difference must not exceed 20%.

#### 2.6.4. Dilution Integrity

The integrity of dilution was determined by adding analyte at a concentration above the ULOQ to blank plasma, which was subsequently diluted with blank plasma at least five times for each dilution factor (2×, 5×, and 10×). The integrity of dilution must match the criteria for accuracy and precision.

#### 2.6.5. Stability

The glipizide bioanalytical stability test was conducted using low (75 ng/mL) and high QC (1250 ng/mL) samples that were tested immediately after preparation and using low (75 ng/mL) and high QC (1250 ng/mL) samples stored under specific conditions and then examined. This stability test includes a short-term stability test, in which the QC samples of glipizide were prepared at room temperature (T0) and frozen in a freezer (−80 °C) for 4 (T4) and 24 h (T24). The QC glipizide samples were stored in a freezer (−80 °C) and thawed at room temperature three times to determine their liquid-freezing stability. Stable QC samples of glipizide ready for injection were stored in the autosampler for 24 h.

### 2.7. Experimental Design for In Vivo Studies

#### 2.7.1. Animals Used in the Study

The UGM Integrated Research and Testing Laboratory provided male Wistar rats weighing between 180 and 250 g and aged 8–10 weeks for the experiments. The rats were maintained in cages (50 cm × 45 cm × 15 cm) with controlled temperature (24 ± 1 °C) and humidity (40–70%) place in a room with an auto-adjusted light cycle of 12 h of bright and 12 h of dark conditions. The feed was 5–10 g/day ABS2, and drinking water (groundwater) was administered ad libitum. The Animal Ethics Committee of Integrated Research and Testing Laboratory UGM, Indonesia authorized the protocol for maintenance and pharmacokinetic study (approval number: 00055/04/LPPT/XII/2021; Yogyakarta, Indonesia).

#### 2.7.2. Sample Preparation

*Andrographis paniculata* (Burm. f.) extract (APE) was obtained using the maceration method with 96% ethanol solvent for 24 h (remaceration two times). Using a rotary evaporator at 50 °C, the filtrate from the maceration was concentrated to get a thick extract. Animals were administered glipizide (GLZ), APE, and andrographolide (AND) standards suspended in 5% sodium carboxy methyl cellulose (CMC-Na).

#### 2.7.3. Pharmacokinetic Interaction Study in Normal and Diabetic Rats

The changes in the pharmacokinetic parameters of glipizide in normal rats were evaluated using pharmacokinetic assays to determine the effects of a 7 day treatment with APE and AND. Twenty animals were used in this pharmacokinetic test and were divided into four groups: CMC-Na or glipizide (5 mg/kg BW) on day 7, a combination of APE (300 mg/kg BW) for 7 days and glipizide (5 mg/kg BW) on day 7, and a combination of AND (15 mg/kg BW) for 7 days and glipizide (5 mg/kg BW) on day 7. The same number of rats in the treatment groups were utilized for the pharmacokinetic evaluation of diabetic rats. The rats were verified to be diabetic (blood glucose values > 200 mg/dL) after being intraperitoneally induced with streptozotocin (STZ) 65 mg/kg BW and nicotinamide 110 mg/kg BW. CMC-Na alone, glipizide (5 mg/kg BW) alone, and combination of APE (300 mg/kg BW) and AND (15 mg/kg BW) with glipizide (5 mg/kg BW) were administered to rats for 28 days. The doses of APE and AND selected relate to optimizing antidiabetic activity, while the dose of glipizide is derived from previous research [40,41].

The blood was collected through the orbital sinus of the rat’s eye (250–300 μL) at 0.25, 0.5, 2, 4, 6, 8, 12, and 14 h after treatment administration to measure the pharmacokinetic parameters. The blood was centrifuged at 4000 rpm for 10 min, and the plasma was kept at −80 °C before analysis. In brief, 200 µL of plasma samples were deprotonated with 1000 µL of acetonitrile. The filtrate and residue were separated by centrifugation at 14,000 rpm and 4 °C for 10 min. The filtrate was successfully treated by deprotonation twice and was dried until no acetonitrile residue remained. The residue was then dissolved in the mobile phase in the HPLC instrument. For the subsequent analysis, 1 mL of the sample solution was placed in an HPLC vial, and the injection volume was set at 20 µL. Glipizide concentrations were determined using an optimized and validated HPLC system. Plasma concentrations of glipizide were measured at each collection time to determine the pharmacokinetic parameters.

### 2.8. Data Analysis

The estimated pharmacokinetic parameters were examined by noncompartmental analysis (NCA) with PKSolver 2.0 USA software. One-way ANOVA was conducted using GraphPad Prism 8 software, and significance was determined at *p* < 0.05 and *p* < 0.01.

## 3. Results

### 3.1. SST

SST parameters such as glipizide bioanalysis retention time, area, asymmetry, and theoretical plate conformed with the EMA guidelines (Figure 1 and Table 1). On the basis of the glipizide SST results in spike plasma, the optimal parameters for the HPLC analytical procedure were obtained. The proposed method can reliably and efficiently quantify glipizide in plasma.

### 3.2. Validation Method

#### 3.2.1. Selectivity

The results for selectivity from six separate samples met the criteria established by the EMA. The interference area on retention time was equivalent to the peak glipizide retention time in blank plasma (≤20% of the glipizide area on the LLOQ). These results demonstrate that this method is selective for analyzing the plasma samples of glipizide.

#### 3.2.2. Calibration Curve and LLOQ

The calibration curve was determined on the basis of the correlation between area (y-axis) and glipizide concentration (x-axis) in the spiked plasma concentration range of 25–1500 ng/mL. Linear regression equations (y = bx + a) were generated using Microsoft Excel 2019. The linearity of the calibration curve was determined using the r-value (correlation coefficient) of 0.999 (R^2^ = 0.998) (Figure 2). Linearity results indicated that the analytical method exhibited a proportionate relationship between the detector response and variations in the analyte concentration. The proposed approach produced a LLOQ of 25 ng/mL. The concentration met the LLOQ criterion, i.e., the acquisition of a percent differentiation value of 20%.

#### 3.2.3. Accuracy and Precision

Accuracy and precision were determined using four analyte concentrations in various plasma spikes, namely, concentrations at LLOQ (25 ng/mL), low QC (75 ng/mL), medium QC (700 ng/mL), and high QC (1250 ng/mL). The results of within-run and between-run accuracy tests matched the requirement: a 15% difference between the average measured concentration of the QC sample and the theoretical concentration (%error), except for the LLOQ at 20% difference. The obtained precision values also met the criteria with a %CV of 15%, except for the LLOQ of 20% (Table 2) and recovery data for LLOQ (80–120%), LQC, medium QC, and high QC (85–115%) (Table 3). On the basis of these two validation parameters, the proposed method can be used for glipizide analysis in plasma samples to obtain a good value for the degree of similarity between the analytical results and the actual analyte concentration, repeatability, and reproducibility.

#### 3.2.4. Dilution Integrity

The dilution integrity test employed the same accurate and precise criteria parameters (%error ≤ 15%, % CV ≤ 15%) to assess the bioanalytical method’s accuracy, precision, and dependability. The findings of dilution integrity test indicated that the analytical method could examine diluted samples with precision and accuracy (Table 4).

#### 3.2.5. Stability

Stability tests were conducted to verify whether the analyte in plasma remains stable and does not degrade throughout bioanalysis and storage. The freeze–thaw stability and short-term stability of glipizide stored at room temperature and −80 °C (4 and 24 h) and in the autosampler (24 h) indicated its good stability results. No significant degradation was observed for the glipizide stored under a variety of conditions (%error ≤ 15 %; % CV ≤ 15%) (Table 5).

### 3.3. Study of Pharmacokinetic Interaction in Normal and Diabetic Rats

Bioanalytical validation using HPLC was successfully applied to determine the several pharmacokinetic parameters of glipizide administered alone and in combination with APE and AND in rat plasma. Figure 3 represents the correlation between glipizide concentration and plasma uptake in normal and diabetic rats for up to 14 h following oral administration. The concentration of glipizide significantly increased in the diabetic rats compared with that in the normal rats. In diabetic rats administered with glipizide in combination with APE, the concentration of glipizide increased. In contrast, when combined with AND, it decreased, altering its pharmacokinetic parameters. Table 6 displays the NCA results for the pharmacokinetic parameters of glipizide.

Co-administration of APE and AND reduced the pharmacokinetic parameters of glipizide such as half-life (T 1/2), area under the first moment curve from zero to infinity (AUMC_0–∞_), and mean residence time (MRT_0–∞_) in the normal group compared with the control (single glipizide administration). In addition, the maximum observed concentration (C_max_) and area under the curve (AUC_0–t_) increased, but the difference was not significant compared with the single glipizide group (*p* > 0.05). Changes in pharmacokinetic parameters were also observed in diabetic rats administered glipizide with APE and AND for 28 days. The C_max_ value increased 2.5 times (*p* < 0.01), and the AUC_0–t_ value increased 1.7 times (*p* < 0.05) in the diabetic rats receiving APE in combination with glipizide compared with those in the diabetic rats receiving glipizide alone. By contrast, the diabetic rats administered with AND exhibited a substantial decrease in various pharmacokinetic parameters, including C_max_ (9 times), AUC_0–t_ (9.6 times), and AUC_0–∞_ (13.6 times) with *p*-values of 0.01.

## 4. Discussion

The glipizide bioanalytical method was validated to establish an appropriate, stable, and dependable quantitative analytical approach for plasma analysis. RP-HPLC/UV was chosen for the analysis of glipizide concentrations in the plasma because of its accuracy and selectivity for bioanalytical studies [42,43]. Bioanalysis necessitates plasma sample pretreatment by protein precipitation to eliminate interfering substances and optimize the sensitivity of the analytical procedure [44,45]. On the basis of the optimization results of the protein precipitation method, acetonitrile was selected because it creates the best plasma matrix devoid of interfering chemicals compared with methanol. Acetonitrile is suitable for usage as a mobile phase in HPLC systems and can attract plasma proteins. Variations in the ratio of mobile phase composition to pH also have an essential impact on determining the analysis method of glipizide in plasma.

The parameters of optimum drug resolution value of endogenous biologic substances, best peak shape, and reasonable retention time were used to determine the composition of the mobile phase. Validation of the percentage differentiation parameter of ≤20% revealed that the developed analytical method acquired a LLOQ of 25 ng/mL, which is comparable with the values in earlier investigations at a concentration of 50 ng/mL [46]. Therefore, the proposed bioanalytic approach can determine analyte concentrations in plasma with accuracy and precision at the LLOQ [47].

Glipizide is a sulfonylurea drug that is more susceptible to thermal degradation than other drugs, such as gliclazide [48,49]. The stability test conducted as part of the validation procedure demonstrated that the changes in glipizide concentrations did not reach statistical significance. Therefore, the proposed technique continued to meet the EMA guidelines. The validated bioanalytical method for glipizide that is selective, accurate, precise, and stable in plasma spikes was then used to analyze the concentration of glipizide co-administered with APE and AND in the plasma of normal and diabetic rats.

Similar to glibenclamide, the elevated concentrations of glipizide in the diabetic group was related to their elevated blood glucose levels [36]. The plasma concentration of glipizide in the diabetic rats was 10 times higher than that in the normal rats. The effects of streptozotocin induction on the gene expression of arachidonic and drug-metabolizing enzymes in the liver of diabetic rats resulted in the elevated concentrations of glipizide [50,51]. These diabetic rats can develop liver diseases that reduce the production and activity of drug-metabolizing enzymes, potentially resulting in pharmacokinetic parameter alterations [52]. Due to micro- and macrovascular alterations, the pathological condition of diabetic rats has been shown to affect the absorption and distribution of a drug. In animal models with diabetes, biotransformation/metabolism and drug excretion are also susceptible to changes [53]. In this study, it was found that differences in the pathological conditions of normal and type 2 diabetic rats (insulin deficiency model) affected the plasma glipizide concentration, thus affecting changes in pharmacokinetic parameters. This insulin-deficient rat model exhibits characteristics of moderated hyperglycemia and is associated with a loss of 60% of the function of β-cells. As a result, the condition of diabetes is reasonably stable [54]. 

The enhanced bioavailability of glipizide in the diabetic rats considerably increased the values of its pharmacokinetic parameters (T 1/2, C_max_, AUC_0–t_, AUC_0–∞_, AUMC_0–∞_, and MRT_0__–∞_). The prolonged T 1/2 value caused glipizide to persist longer in the diabetic rats than in the normal rats. Owing to the decreased ability of the CYP2C9 enzyme to generate the inactive metabolites of glipizide, the elevation in glipizide’s pharmacokinetic parameters could affect the potential pharmacological activity of this drug. When glipizide was combined with natural compounds that induce or inhibit the activity of metabolizing enzymes, the changes in enzyme activity resulted in pharmacokinetic HDIs in the diabetic rats. APE enhanced the pharmacokinetic parameters (C_max_ and AUC_0–t_) of glipizide in normal and diabetic rats. The increase in these two pharmacokinetic parameters indicated an increase in plasma glipizide concentration (bioavailability) [55]. In the diabetic rats, the increased bioavailability of glipizide due to its combination with APE had a beneficial effect, i.e., pharmacological enhancement. APE exhibits glucose-lowering efficacy in rats by increasing the mRNA and protein expression of GLUT-4 and insulin expression in pancreatic beta cells [56,57]. A study reported evidence of an increase in the pharmacological action of the medicine (synergistic effect) when the anti-inflammatory agents etoricoxib and naproxen were combined with APE [58].

The additional adjustment in the bioavailability of glipizide in diabetic rats was hypothesized to be the result of APE’s metabolizing enzyme’s inhibitory impact. In vivo studies and human/rat hepatocyte culture experiments revealed that APE can significantly inhibit CYP2C9 and CYP3A4 expression [25]. Similar to antidiabetic drug sulfonylureas, the APE-induced increase in bioavailability also occurred for gliclazide, which is also metabolized by CYP2C9 [32]. AND, the major secondary metabolite of *A. paniculata*, dramatically reduced the bioavailability of glipizide in diabetic rats (decreased parameters C_max_, AUC_0–t_, AUC _0–∞_, and AUMC _0–∞_) but showed the opposite effect on normal mice (nonsignificant increase in bioavailability). The decrease in the plasma concentration of glipizide might be related to its slow absorption due to co-administration with AND (marked by a long T_max_ value). Tolbutamide, an antidiabetic, similarly experiences a reduction in pharmacokinetic drug bioavailability when combined with AND. AND enhances the gene transcription and enzyme activity of CYP1A1/2, CYP2C6/11, and CYP3A1/2 by activating AhR and binding to PXR in the cell nucleus and by significantly increasing CYP2C activity. In addition, this compound induces the expression and activity of enzymes to increase the metabolism of tolbutamide and inhibit its deposition to the target of action [59]. Owing to its interaction with AND, the bioavailability of nabumetone decreases and negatively affects its pharmacological activity. A significant reduction in the pharmacokinetic parameters (C_max_, T_max_, and AUC_0–t_) and antiarthritic efficacy of nabumetone was observed when it was administered with AND [28].

Owing to the inconsistency between the mechanism of HDIs during the pharmacokinetic phase and the pharmacological activity, additional interaction studies are warranted to determine the effect of APE and AND on the antidiabetic pharmacological activity of glipizide. Different mechanism pathways exist for HDIs in the pharmacokinetic and pharmacodynamic phases; pharmacokinetics focuses on interactions in the ADME phase of pharmaceuticals, and pharmacology is mainly concerned with the synergistic, additive, or antagonistic effects of HDIs [11,60]. Changes in drug bioavailability in plasma affect the binding activity of the medication to its target of the action, although the two mechanisms utilize distinct pathways. Therefore, the results of this study can be applied in the therapeutic evaluation of HDIs affecting the efficacy of diabetes treatment. The limitation of this study is that the pharmacokinetic data in the animal model are insufficient to confirm the existence of herb–drug interactions; thus, it is necessary to extrapolate the data in humans to assess clinical significance.

## 5. Conclusions

A straightforward and rapid bioanalysis method for measuring glipizide in plasma was developed. Validation revealed that the technique exhibits accuracy, precision, selectivity, and sensitivity, and linearity (*r* = 0.999) within the concentration range of 25–1500 ng/mL. Glipizide remained stable in the plasma of normal and diabetic rats. The analytical approach was then utilized to examine the pharmacokinetics of HDIs. APE administration significantly altered the pharmacokinetic parameters (C_max_ and AUC) of glipizide (*p* < 0.05), thus increasing the bioavailability of glipizide. AND administration significantly decreased (*p* < 0.05) the parameters (T 1/2, C_max_, and AUC) of glipizide in diabetic rats. APE and AND that are co-administered with glipizide are a source of potential herb–drug interactions. Although the effect on antidiabetic activity needs to be studied further, this research can reflect the concern in the combination of herbal use for diabetes therapy.

## Figures and Tables

**Figure 1 molecules-27-06901-f001:**
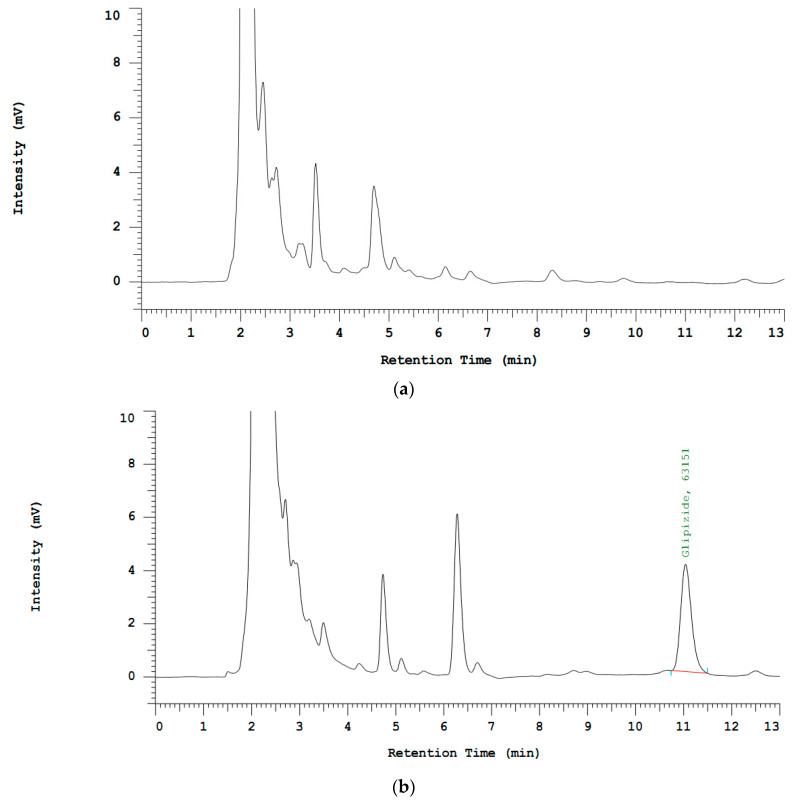
HPLC chromatogram of blank plasma (**a**) and glipizide (retention time 11.07 min) in plasma (**b**) using mobile phase KH_2_PO_4_, pH 4.25, acetonitrile (57.43%), and flow rate 1 mL/min.

**Figure 2 molecules-27-06901-f002:**
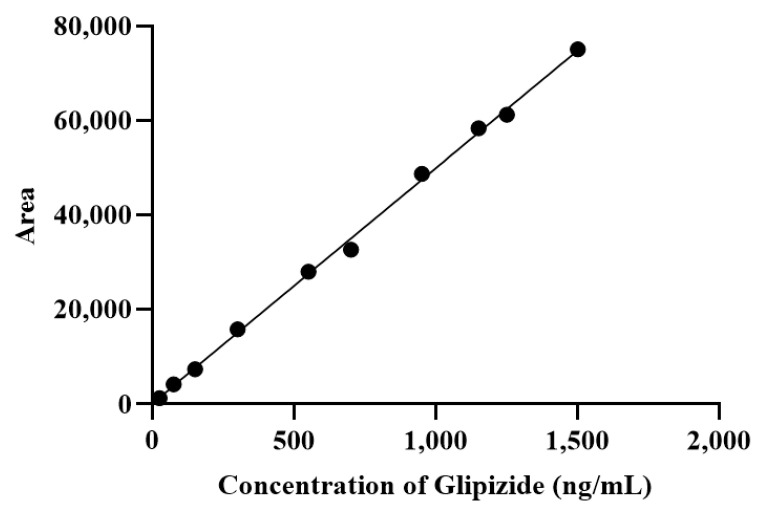
Calibration curves for glipizide using HPLC to determine the relationship between concentration and peak area. The equation obtained was y = 49.858x + 143.83 (R^2^ = 0.9983), F value = 4812.72.

**Figure 3 molecules-27-06901-f003:**
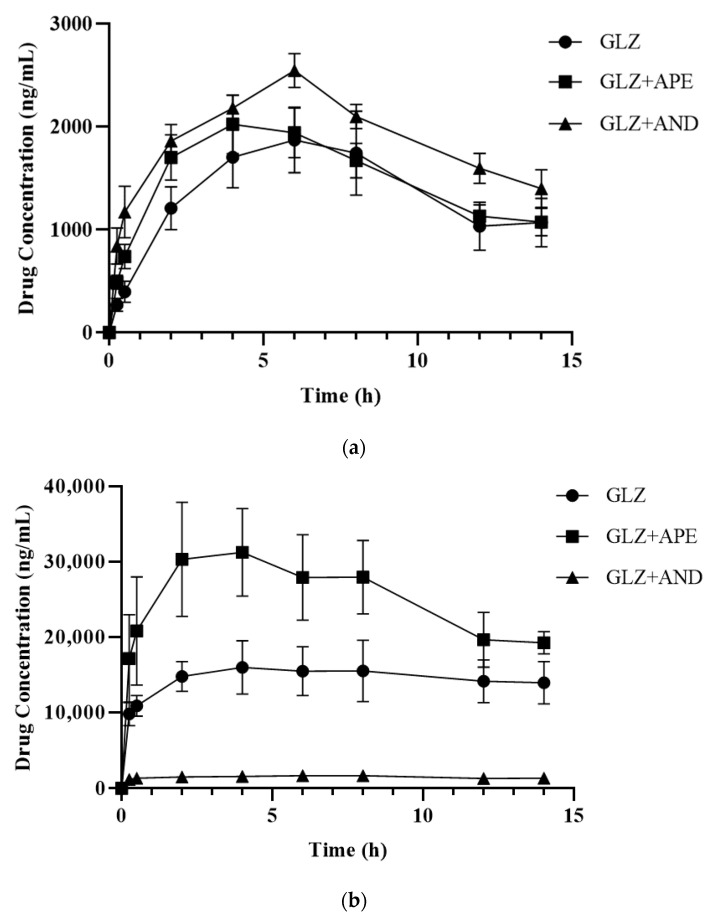
Concentrations of glipizide in the plasma (ng/mL ± S.E.M.) of normal (**a**) and diabetic (**b**) rats treated with glipizide (GLZ 5 mg/kg BW) alone or in combination with *Andrographis paniculata* extract (APE 300 mg/kg BW) and andrographolide (AND 15 mg/kg BW).

**Table 1 molecules-27-06901-t001:** System suitability test (SST) results of glipizide in spike plasma.

Parameters	Result	Acceptance Criteria
Retention time	0.30	RSD ≤ 2.0%
Area	0.36	RSD ≤ 2.0%
Tailing factor (asymmetry)	1.23	≤2.0
Number of theoretical plates	11,336	≥2000

**Table 2 molecules-27-06901-t002:** Results of glipizide accuracy and precision in spike plasma.

Concentration (ng/mL)	% Error				% CV			
	Within (*n* = 6)			Between (*n* = 18)	Within (*n* = 6)			Between (*n* = 18)
	Day 1	Day 2	Day 3		Day 1	Day 2	Day 3	
25	−2.89	0.54	10.64	2.76	10.82	5.69	3.89	7.36
75	−1.53	3.95	8.84	3.75	6.43	5.08	3.84	5.12
700	−1.65	9.56	−4.17	1.25	5.55	3.92	3.82	4.43
1250	9.20	13.30	5.34	9.28	5.13	0.34	6.08	3.85

**Table 3 molecules-27-06901-t003:** Recovery data for glipizide in spiked plasma (*n* = 18).

Concentration (ng/mL)	Measured Concentration (ng/mL) (Mean ± SD)	% Recovery (±CV)
25	25.69 ± 1.76	102.76 ± 0.07
75	77.81 ± 3.89	103.42 ± 0.05
700	708.72 ± 5.14	101.29 ± 0.07
1250	1366 ± 49.75	109.28 ± 0.04

**Table 4 molecules-27-06901-t004:** Uji Dilution integrity glipizide in spike plasma.

Dilution Factor	Mean ± SD Measurable Conc. (ng/mL)	Accuracy (% Error)	Precision (% CV)
2×	199.04 ± 10.94	−0.48	5.50
5×	423.20 ± 24.54	5.80	5.80
10×	1054.29 ± 74.57	5.43	7.07

**Table 5 molecules-27-06901-t005:** Stability test of glipizide in spike plasma.

Stability Study		Mean ± SD Measurable Conc. (ng/mL)	Accuracy (% Error)	Precision (% CV)
Short-Term Stability				
T0	Low QC 75 µ/mL High QC 1250 µ/mL	81.10 ± 2.06	8.13	2.54
		1239.79 ± 68.22	−0.82	5.50
T4		69.99 ± 1.61	−6.68	2.30
		1285.25 ± 10.84	2.82	0.84
T24		70.30 ± 2.90	−6.27	4.12
		1311.63 ± 30.52	4.93	2.33
Freeze–thaw stability		78.65 ± 4.84	4.86	6.15
		1230.17 ± 112.55	−1.59	9.15
Autosampler		76.37 ± 2.27	1.83	2.98
		1233.47 ± 71.06	−1.32	5.76

**Table 6 molecules-27-06901-t006:** Pharmacokinetic parameters of glipizide in the plasma of normal and diabetic rats.

Parameters	Unit	Value (±SEM)					
		Normal Rats			Diabetic Rats		
		GLZ	GLZ + APE	GLZ + AND	GLZ	GLZ + APE	GLZ + AND
**T 1/2**	h	11.7 ± 4.19	7.07 ± 0.49	6.8 ± 0.6	39.06 ± 12.4	13.5 ± 3.9	23.59 ± 8.3
Tmax	h	5.6 ± 0.7	4.4 ± 0.4	6 ± 0	5.2 ± 1.3	3.7 ± 1.2	6.4 ± 0.75
Cmax	ng/mL	2038.3 ± 359.4	2134 ± 233.2	2545.6 ± 165	17,049.6 ± 3407	44,634.7 ± 5.471 **	1876.4 ± 133.2 *
AUC_0–t_	ng/mL·h	25,130.3 ± 5.228	28,231.4 ± 2978	35,893 ± 3048	204,126.9 ± 41,879	356,362.8 ± 37.946 *	21,052.9 ± 1.292 **
AUC_0–∞_	ng/mL·h	34.266 ± 3379	31,729.2 ± 3120	41,127.7 ± 4339	978,757.3 ± 279,871	734,414.7 ± 123.596 *	71.719 ± 22.203 **
AUMC_0–∞_	ng/mL·h^2^	637,882.6 ± 162,429	378,090.9 ± 29,818	522,329 ± 89,356	74,019,560 ± 3,708,6791	17,673,567.5 ± 7,373,051	3,632,224 ± 2,286,205
MRT_0–∞_	h	19.3 ± 6.1	12 ± 0.4	12.4 ± 0.89	57.8 ± 17.8	20.9 ± 5	35.8 ± 12.04
**Cl/F**	(ng)/(ng/mL)	2.662 ± 1.1381	1.713 ± 0.30	1.22 ± 0.09	0.3067 ± 0.059	0.1263 ± 0.02006	2.272 ± 0.179
**Vd/F**	(ng)/(ng/mL)/h	0.152 ± 0.0159	0.165 ± 0.02	0.13 ± 0.02	0.0075 ± 0.002	0.0076 ± 0.00118	0.094 ± 0.023

All values are expressed as the mean ± SEM. GLZ = glipizide; APE = *Andrographis paniculata* extract; AND = andrographolide. * Significant at *p* < 0.05; ** significant at *p* < 0.01, when compared to a single glipizide treatment. T 1/2 = half-life; Tmax = Maximum observed time; Cmax = maximum observed concentration; AUC_0–t_ = area under the curve from zero to time t; AUC_0–∞_ = area under the curve from zero to infinity; AUMC_0–∞_ = area under the first moment curve from zero to infinity; MRT = mean residence time from zero to infinity; Cl/F = clearance; Vd/F = volume of distribution.

## Data Availability

The data described in this study are accessible from the author by request.

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
