# Peer review of "Pharmacokinetic Herb-Drug Interactions of Glipizide with Andrographis paniculata (Burm. f.) and Andrographolide in Normal and Diabetic Rats by Validated HPLC Method"

_molecules, 2022, doi:10.3390/molecules27206901_

Round 1

Reviewer 1 Report

This manuscript described a validated method for quantitation of glipizide in plasma samples by HPLC-UV method. Further the assay was applied for its pharmacokinetic interaction study with ethanolic extract of Andrographis paniculata (Burm.f.) and andrographolide (AND) in rats. The novelty of the study is limited as method development point of view as many methods already reported. Author should more emphasize the interaction studies and it is interesting to see the herb drug Interaction in both normal and diabetic rats. Following points need to be address before its consideration to publication.

1.       Title should need to be change and more emphasize to integration studies.  Like “Pharmacokinetic interaction study of glipizide with Andrographis paniculata (Burm.f.) and andrographolide in normal and diabetic rats by validated HPLC method.

2.       Section 2.4. During protein precipitation method, sample drying and reconstitution is performed with the low volume of solvent to increase the sensitivity of the assay. In this method, 200 uL sample was taken and reconstitution performed with 1 ml of mobile phase. Why it was not performed with low volume of mobile phase e.g .200 ul. Explain?

3.       Section 2.6.2. Kindly check the calibration curve concentration is it in ug/mL?

4.       Recovery study of validation parameter is missing. It should be performed for new method and required to be consistent throughout the QC samples and concentration independent.

5.       Section 2.7.2. How the dose of glipizide and herval products were decided? Kindly provide the administered dose reference. Or justify it.

6.       The blank HPLC chromatogram is missing. It should be provided for a new method to ensure the selectivity of the assay.

7.       From Fig. 2 it is looking that no internal standard was used for this assay. For any bioanalytical assay IS must be including to compensate the sample lost during extraction procedure. Moreover, it is mandatory to do recover study to prove that it is 100 % reproducible.

8.       Author should mention the significance of this study in conclusion and further any recommendation for the dose adjustment or any clinical study warranted.

9.        Since many methods for glipizide already reported, therefore authors should compare the novelty of this assay in compared to other.

This manuscript described a validated method for quantitation of glipizide in plasma samples by HPLC-UV method. Further the assay was applied for its pharmacokinetic interaction study with ethanolic extract of Andrographis paniculata (Burm.f.) and andrographolide (AND) in rats. The novelty of the study is limited as method development point of view as many methods already reported. Author should more emphasize the interaction studies and it is interesting to see the herb drug Interaction in both normal and diabetic rats. Following points need to be address before its consideration to publication.

1.       Title should need to be change and more emphasize to integration studies.  Like “Pharmacokinetic interaction study of glipizide with Andrographis paniculata (Burm.f.) and andrographolide in normal and diabetic rats by validated HPLC method.

2.       Section 2.4. During protein precipitation method, sample drying and reconstitution is performed with the low volume of solvent to increase the sensitivity of the assay. In this method, 200 uL sample was taken and reconstitution performed with 1 ml of mobile phase. Why it was not performed with low volume of mobile phase e.g .200 ul. Explain?

3.       Section 2.6.2. Kindly check the calibration curve concentration is it in ug/mL?

4.       Recovery study of validation parameter is missing. It should be performed for new method and required to be consistent throughout the QC samples and concentration independent.

5.       Section 2.7.2. How the dose of glipizide and herval products were decided? Kindly provide the administered dose reference. Or justify it.

6.       The blank HPLC chromatogram is missing. It should be provided for a new method to ensure the selectivity of the assay.

7.       From Fig. 2 it is looking that no internal standard was used for this assay. For any bioanalytical assay IS must be including to compensate the sample lost during extraction procedure. Moreover, it is mandatory to do recover study to prove that it is 100 % reproducible.

8.       Author should mention the significance of this study in conclusion and further any recommendation for the dose adjustment or any clinical study warranted.

9.        Since many methods for glipizide already reported, therefore authors should compare the novelty of this assay in compared to other.

Reviewer 2 Report

This article is well-designed and it is a valuable addition to the literature. However, there are several aspects, which need clarification and extension. My specific comments are listed below:

1. The Authors stated (section 1, line 92) that RP-HPLC/UV method is simpler, more sensitive, and less expensive compared to LC-MS/MS method. While I can agree with the statement that it is less expensive than MS / MS methods, a number of studies documented that the lower limit of quantitation of glipizide with MS detection was 1.00 ng/mL. Please refer to the published data confirming your opinion.

2. Streptozotocin (STZ) is a highly selective pancreatic islet β-cell-cytotoxic agent that is often administered to cause diabetes. Have the authors used or developed STZ model of type 2 diabetes in rats with nicotinamide to partially protect the β-cells against STZ, to be sure that produces a model of insulin-deficient (type 2), but not insulin-resistant (type 1)? Whether the type of diabetes induced was taken into account? Could the type of diabetes affect the glipizide concentration? Please comment your model choice in paragraph 2.7.2 or the Discussion section.

3. In section 2.7.2, line 201 the Authors report a volume of 250-300 mL. The value is surprisingly large, is it really correct? Please explain if this is the volume of a single blood sample taken from a rat.

In this same section, I recommend adding the sample size of injection to the chromatographic system.

4. Were APE and AND administered to rats in the form of extracts? How were they obtained? Whether they were ready-made standards. Were they administered to animals  in the form of prepared solutions? This information should be listed in section 2.7.2 for clarification.

5. Whether the equation y = bx + a was used for further calculations despite the statement the intercept value of the standard curve was not significant?

6. The correlation coefficient (r) is commonly used to evaluate the degree of linear association between two variables. However, r is not an appropriate measure for linearity and should be evaluated by statistical methods. Two statistical tests, including the Lack-of-fit (LOF) and Mandel’s fitting tests, are suitable for the validation of the linear calibration model. Have the authors analyzed any of the mentioned tests?

7. The following parameters were estimated by noncompartmental analysis: T0,5, Tmax, Cmax, AUC, AUMC, and MRT. It would also be worthwhile to additionally determine the clearance and volume of distribution.

8. In section 3.3, line 283, the authors report that in rats with GLP +APE and GLZ +AND, the concentration of GLZ increased, although the results shown in Figure 3 and Table 5 indicate that in diabetic rats administered GLZ+AND, glipizide levels decrease. Please clarify this issue

9. The authors should underline the limitations of the study.

Reviewer 3 Report

This manuscript established a straightforward and rapid bioanalysis method for measuring glipizide (GLZ) in rat plasma which exhibited accuracy, precision, selectivity, and sensitivity, and linearity (r = 0.999) within the concentration range of 25-1500 ng/mL. This method was then utilized to examine the pharmacokinetics of herb-drug interactions (HDIs) between GLZ and APE (Andrographis paniculata ethanolic extract) or AND (andrographolide) in rat plasma.

Overall, this manuscript is logically organized and suited for publication in molecules after addressing the following points.

1.      According to the introduction, AND can inhibit the kinetics of several enzymes, including CYP2E1, and decrease the expression of CYP2C9 and CYP3A76, but in the discussion, AND also enhances the gene transcription and enzyme activity of CYP1A1/2, CYP2C6/11, and CYP3A1/2. What is the real cause of the decrease in the plasma concentration of glipizide combination with AND?

2.      It is recommended to put the concentrations of glipizide in the diabetic and normal rats in one figure to make the results clearer.

3.      While the English in the manuscript is not too bad, there are some typos mistakes. It will be the authors’ responsibility to polish the manuscript beyond the suggested corrections:

a. In abstract, “GLZ was administered alone and in combination with APE and AND to normal and diabetic rats.” An “or” is better than an “and” here, since APE and AND are used respectively.

b. Lines 34 and 35, “ AUC0–t” should be “ AUC0–t”; line 348, “AUC0-” should be “AUC0-∞”; line 371, “AUC 0-, and AUMC 0-” should be “AUC 0-∞, and AUMC 0-∞”.

c. Line 246, “(R2 = 0.9983)” should be “(R2 = 0.9983)”.

d. In Table 1, the author should check the underlines for “Acceptance criteria”; same question for line 232.

e. In Table 5, are the values correct for “GLZ + AND in Diabetic rats”? Why the AUC0–t increased when T 1/2 and Cmax both decreased?

Round 2

Reviewer 1 Report

Authors responses are satisfactory and can be consider for publication in molecules. Only need to change the title again as suggested below.

Pharmacokinetic Herb-Drug Interactions of glipizide with Andrographis paniculata (Burm.f.) and andrographolide in normal and diabetic rats by validated HPLC method
